# Effects of Compound Mycotoxin Detoxifier on Alleviating Aflatoxin B_1_-Induced Inflammatory Responses in Intestine, Liver and Kidney of Broilers

**DOI:** 10.3390/toxins14100665

**Published:** 2022-09-24

**Authors:** Hongwei Guo, Ping Wang, Chaoqi Liu, Ting Zhou, Juan Chang, Qingqiang Yin, Lijun Wang, Sanjun Jin, Qun Zhu, Fushan Lu

**Affiliations:** 1College of Biology and Food Engineering, Huanghuai University, Zhumadian 463000, China; 2College of Animal Science and Technology, Henan Agricultural University, Zhengzhou 450046, China; 3Guelph Research and Development Centre, Agriculture and Agri-Food Canada, Guelph, ON N1G 5C9, Canada; 4Henan Delin Biological Product Co., Ltd., Xinxiang 453000, China; 5Henan Puai Feed Co., Ltd., Zhoukou 466000, China

**Keywords:** broilers, compound mycotoxin detoxifier, aflatoxin B_1_, inflammatory response, tissue damage

## Abstract

In order to alleviate the toxic effects of aflatoxins B_1_ (AFB_1_) on inflammatory responses in the intestine, liver, and kidney of broilers, the aflatoxin B_1_-degrading enzyme, montmorillonite, and compound probiotics were selected and combined to make a triple-action compound mycotoxin detoxifier (CMD). The feeding experiment was divided into two stages. In the early feeding stage (1–21 day), a total of 200 one-day-old Ross broilers were randomly divided into four groups; in the later feeding stage (22–42 day), 160 broilers aged at 22 days were assigned to four groups: Group A: basal diet (4.31 μg/kg AFB_1_); Group B: basal diet with 40 μg/kg AFB_1_; Group C: Group A plus 1.5 g/kg CMD; Group D: Group B plus 1.5 g/kg CMD. After the feeding experiment, the intestine, liver, and kidney tissues of the broilers were selected to investigate the molecular mechanism for CMD to alleviate the tissue damages. Analyses of mRNA abundances and western blotting (WB) of inflammatory factors, as well as immunohistochemical (IHC) staining of intestine, liver, and kidney tissues showed that AFB_1_ aggravated the inflammatory responses through NF-κB and TN-α signaling pathways via TLR pattern receptors, while the addition of CMD significantly inhibited the inflammatory responses. Phylogenetic investigation showed that AFB_1_ significantly increased interleukin-1 receptor-associated kinase (IRAK-1) and mitogen-activated protein kinase (MAPK) activities (*p* < 0.05), which were restored to normal levels by CMD addition, indicating that CMD could alleviate cell inflammatory damages induced by AFB_1_.

## 1. Introduction

Aflatoxins (AFs) produced by *Aspergillus flavus* and *Aspergillus parasiticus* constitute a family classified as difuranocoumarins, which are highly substituted coumarin derivatives containing a fused dihydrofurofuran moiety [1,2]. At present, more than 20 kinds of AFs have been isolated and identified, among which AFB_1_ holds the highest toxicity, followed by AFG_1_, AFB_2_, and AFG_2_. AFB_1_ has been classified as a human class I carcinogen by the International Agency for Cancer Research [3]. Poultry is more sensitive to AFB_1_ than the other kinds of animals, and AFB_1_ residues in poultry cause potential hazards for human health [4]. AFB_1_ has been known to contaminate animal feedstuffs and diets. The antioxidant capacity and immunity would be weakened [5] and the internal organs (especially the liver) would be damaged [6] once the contaminated feed was ingested by the broilers.

The liver is the primary organ attacked by AFB_1_, and AFB_1_-induced liver cell cytotoxicity is involved in the interactions of inflammation, oxidative stress, and liver metabolic enzymes. AFB_1_ can be catalyzed by liver phase I metabolic enzyme (cytochromes P450) in the mitochondrial to form AFB_1_-exo-8,9-epoxide (AFBO), which can bind to DNA and inhibit DNA replication and protein expression to cause the damage and cancelation of hepatocytes [7]. AFB_1_ stimulates cell membrane phospholipid A2 to cause lipid peroxidation, and induces the production of reactive oxygen free radicals (ROS), which could induce DNA to form 8-OHdG resulting in cell damage [8,9]. Low doses of ROS can be neutralized by phase 2 metabolic enzyme (glutathione S-transferases); excessive ROS could induce the endoplasmic reticulum oxidative stress, the increase of inflammatory factors, mitochondrial apoptosis, and other reactions [10]. The cytokines exist as markers of inflammatory response during the whole cell injury cycle [11]. These cell damages were finally reflected in the changes of physical and chemical indexes of broilers.

Biological detoxifiers such as beneficial microorganisms [12,13] and enzymes [14] have been proven to degrade AFB_1_ in vitro and regulate inflammation and oxidative stress in vivo [13]; therefore, biological detoxifiers have good effects on alleviating AFB_1_-induced damage in broiler production [15,16]. Physical adsorbents such as montmorillonite and bentonite can absorb AFB_1_ to alleviate its toxic effect on broiler production [17]. However, the effective addition dose of physical adsorbent is 0.2–0.3% in animal diets, and this high-dose physical adsorbent addition may cause nutrients (amino acids, minerals, and vitamins) to also be adsorbed, which results in nutrient deficiency [18]. Furthermore, it was reported that AFB_1_-adsorption effectiveness of montmorillonite was greatly affected by pH value, which means that the adsorbed mycotoxin can lead to desorption in the animal gut and cause secondary contamination [19]. In general, physical adsorbents (montmorillonite) and biological agents (probiotics, mycotoxin-degradation enzymes) exhibit different relative merits for mycotoxin degradation or removal in vitro and in vivo, and the combination of physical and biological methods may be more effective for AFB_1_ degradation [20,21].

In the previous research in our laboratory, the artificial gastrointestinal fluid was used to screen out the compound mycotoxin detoxifier (CMD) to degrade AFB_1_ effectively, in which it contained aflatoxin B_1_-degrading enzyme, montmorillonite, and compound probiotics [22]. The effect of CMD on alleviating aflatoxin B_1_-induced cytotoxicity in the chicken embryo primary intestinal epithelium, liver, and kidney cells has been evaluated in vitro [23]. To further clarify the effectiveness of CMD in vivo, this study will explore the mechanism of CMD in alleviating inflammation responses in the intestine, liver, and kidney of broilers caused by AFB_1_.

## 2. Results

### 2.1. Effect of CMD on mRNA Abundances of Some Genes in Intestinal, Liver and Kidney Tissues of Broilers

Figure 1 showed that the expressions of seven genes were significantly up-regulated in group B (*p <* 0.05); however, most of them, except for IL-6 and IL-8 in the intestinal tissue, were down-regulated with the addition of CMD in group D (*p <* 0.05). Compared to the control group, a significant increase of NOD1 expression and a remarkable decrease of NF-κBp65 expression in the intestinal tissue and kidney tissue was found in group C (*p <* 0.05). Compared to the control group, the gene expressions of TLR2, NF-κBp65, iNOS, TNF-α, and IL-8 in the kidney tissue of group B were significantly increased (*p <* 0.05); however, the up-regulated genes were all down-regulated with the addition of CMD in group D (*p >* 0.05). Compared to the control group, TLR2 and IL-6 expressions in kidney tissue of group C were significantly decreased (*p <* 0.05). 

### 2.2. Effect of CMD on Expression Levels of NF-κB, TLR and NOD in Liver Tissue by WB Analysis

The protein expressions of NF-κB, TNF-α, and TLR in liver tissues were shown in Figure 2. Compared to the control group, the protein expressions of NF-κB and TLR in Group B were significantly up-regulated (*p <* 0.05); however, both of them were significantly down-regulated by CMD addition. The protein expression of NF-κb in Group C was significantly decreased, compared with the control group (*p <* 0.05). There was no significant difference for NOD among the four groups in liver tissues (*p* > 0.05).

### 2.3. Effect of CMD on Expression Levels of Caspase-3 in Intestinal, Liver and Kidney Tissues by IHC Analysis

The Caspase-3 immunoreactivity in jejunum, liver, and kidney tissues were presented in Figure 3, and the PRC and COD results were presented in Table 1. In group B, high PRC and COD were observed in the intestine and kidney (grade 3) and the liver (grade 2), which showed obviously brown nuclei. Compared to Group B, the PRC and COD were significantly reduced in three kinds of tissue by CMD addition (*p <* 0.05). The AFB_1_-induced tissue damage degrees calculated by PRC were: liver > kidney > intestine (*p* > 0.05); the protective effects of CMD on alleviating AFB_1_-induced tissue damage calculated by COD were the same as above, i.e., liver > kidney > intestine (*p* > 0.05).

### 2.4. Effect of CMD on Expression Levels of TNF-α in Intestinal, Liver and Kidney Tissues by IHC Analysis

The TNF-α immunoreactivity in jejunum, liver, and kidney tissues were presented in Figure 4, and the PRC and COD results were presented in Table 2. The PRC and COD indexes showed the same trend among four groups. COD and PRC in Group B were significantly increased (*p <* 0.05), but were decreased to the same level as the control group by CMD addition (*p* > 0.05). AFB_1_-induced damage degrees of tissues calculated by PRC were: liver > intestine > kidney (*p* > 0.05), while the protective effects of CMD calculated by COD were: kidney > liver > intestine.

### 2.5. Effect of CMD on Expression Levels of NF-κB in Intestinal, Liver and Kidney Tissues by IHC Analysis

The NF-κB immunoreactivity in jejunum, liver, and kidney tissues were presented in Figure 5, and the PRC and COD results were presented in Table 3. The PRC and COD indexes showed the same trend among the four groups. COD and PRC in group B were significantly increased (*p <* 0.05), but were decreased to the same level as the control group by CMD addition (*p* > 0.05). AFB_1_-induced tissue damage degrees calculated by PRC were: intestine > liver > kidney (*p* > 0.05), while the protective effects of CMD calculated by COD were: kidney > intestine > liver (*p* > 0.05).

### 2.6. Gut Microbial Community Influenced by CMD and AFB_1_

The microbiota compositions in jejunum content at the genus level were presented in Figure 6. Compared with the control group, the abundances of *Staphylococcus* and *Escherichia-Shigella* were significantly increased in Group B, and the abundances of *Lactobacillus*, *Burkholderia-caballeronia-paraburkholderia*, *Romboutsia,* and *Corynebacterium* were significantly decreased (*p <* 0.05). However, these changes were returned to almost the same levels as the control group by CMD addition in Group D (*p* > 0.05). The abundance of *Lactobacillus* in group C was significantly higher than in the other three groups, which indicates that AFB_1_ could disturb the gut microbiota and CMD addition could keep gut microbiota stable.

### 2.7. Effect of CMD on Expression Levels of TNF-α in Intestinal, Liver and Kidney Tissues by IHC Analysis

In order to further explore the interaction between intestinal microbes and inflammation, kinases related to TNF-α, NFKB, TLR, and NOD signaling pathways were selected for collation and analysis. The results showed that only three kinases based on PICRUSt function prediction were significantly changed (Table 4). Compared to the control group, the function enrichment of IRAK-1 in Group B was significantly increased, and the function enrichment of JAK was significantly decreased (*p <* 0.05); however, the functions enrichment of JAK and MAPK in Group C were significantly increased, and the function enrichment of IRAK-1 was significantly decreased (*p <* 0.05). Compared to Group B, the function enrichment of IRAK-1 was significantly decreased (*p <* 0.05), and the function enrichment of JAK was significantly increased with the addition of CMD in Group D (*p <* 0.05).

## 3. Discussion

There is a general agreement that dietary aflatoxins reduce weight gain, feed conversion rate, and increase tissue and organ damage for broilers [24]. Adding mycotoxin detoxification agents is the most widely used method for detoxification in the mycotoxin- contaminated diets. Previous studies showed that probiotics [25], montmorillonite [17], and the combination of probiotics and mycotoxin-degradation enzymes [16] could reduce mycotoxin toxicity for broilers. Although adding antidotes significantly alleviates the direct damage of mycotoxin to animals, the potential damages, such as immune suppression, oxidative stress, and inflammatory inhibition, still exist widely in broiler production [26]. The inflammatory response marked by cytokine secretion runs through the entire cycle of cell injury [7]. Toxins can up-regulate the expression of pro-inflammatory factors such as IL-6, IL-8, TNF-α, iNOS, and NF-κB signaling pathways through a variety of inflammatory pathways, which is consistent with the results of the present study. It was reported that toxins were used to stimulate porcine intestinal epithelial cells, leading to a significantly increased expression of IL-6 [27]. Another report showed that the levels of IL-6, IL-1, and TNF-α in serum were significantly increased, while the IL-10 level was decreased when the mice were fed with an AFB_1_-contamined diet [28]. It was reported that iNOS is related to many diseases and the regulation of various molecules such as NF-κB, AP-1, IRF, and NF-IL6 [29]. In addition, the expression of iNOS is related to the activation of cell signaling proteins such as PI3K, protein kinase, JAK-2, and mitogen-activated protein kinase [30]. 

In the present experiment, the expressions of iNOS in three kinds of tissues were consistent with the trend of NF-κBP65. It was inferred that AFB_1_ can up-regulate iNOS expression through the NF-κB pathway. The different expressions in different tissues may be due to their different sensitivity to mycotoxins. In a previous study, we used AFB_1_ to stimulate broiler intestinal, liver, and kidney cells, and found that liver and kidney cells are significantly more sensitive than intestinal cells [23]. The activation of NF-κB can promote the up-regulation of a variety of pro-inflammatory factors (IL-6, TNF-α) and chemokines, and even cause cell apoptosis [31]. It was reported that ileum injury induced by AFB_1_ increased the production of AFB_1_-DNA adducts by upregulating the expressions of CYP1A1 and CYP1A2 and increased DNA damage and oxidative stress via the Nrf2/ Keap1 and NF-κB/NLRP3 signaling pathways in ducks [14]. 

Pattern recognition receptors (PRRs) play an important role in innate immunity to be considered as the first line of defense against microbial infections, in which inflammatory pathways will be activated by PRRs during the early stages of the exogenous infection [32,33]. At present, studies on the effects of AFB_1_ on PRRs mainly focus on Toll-like receptors (TLRS) and nucleotide binding oligomerization domain-like receptors (NLRs). It was reported that AFB_1_ evidently decreased mRNA expressions of TLR2, TLR4, and TLR7 in the small intestine of broilers [34]. We used different concentrations of AFB_1_ to stimulate primary chicken embryo intestinal cells (100 μg/mL) and liver and kidney cells (40 μg/mL) for 12 h, and the results showed that TLR2 was significantly up-regulated in kidney cells, while there was no significant difference in other cells [23]. The previous research also showed that AFB_1_ can induce oxidative stress and activate the NF-κB signaling pathway by activating NOD-like receptors (NLRs) [35,36]. Yan et al. showed that AFB_1_ dose-dependently activated the NLRP3 signaling pathway and NF-κB inflammatory pathway in rat cardiac tissue [36]. In this study, the NOD expression trend was not completely consistent in the three kinds of tissues, which may be caused by the different cell sensitivities to AFB_1_ [23].

According to the Figure 3, Figure 4 and Figure 5, microscopic damages by AFB_1_ were clearly observed. The damages of AFB_1_ to the intestinal tract mainly present barrier function loss and inflammatory reaction, lymphocyte or monocyte infiltration, mucosal hyperplasia, vacuolar degeneration, and even decrease intestinal villus height and increase intestinal crypt depth [37,38]. In this study, disruption of the intestinal villi was clearly observed, while the disruption did not decrease with the addition of CMD. It suggested that the AFB_1_-induced negative effect in villus was irreversible and CMD did not significantly reduce the direct intestinal damage caused by AFB_1_.

The liver is the primary organ attacked by AFB_1_, which can cause many microscopic damages including high-level eosinophile granulocyte and monocytes, lipid vacuoles [39], inflammatory cell proliferation and infiltration, edema, and hepatocytes degeneration [40], in agreement with this study. However, the CMD additions could alleviate mycotoxin negative effects on liver tissue damage. The results of this experiment showed that the CMD could significantly alleviate liver microcosmic damage caused by AFB_1_. It was reported that compound probiotics with aflatoxin B_1_-degrading enzymes could improve AFB_1_ metabolism, hepatic cell structure, and antioxidant activity of broilers exposed to AFB_1_-contaminated diets [41]. Another report demonstrate that the gut microbiome acts at a distance to activate host antioxidant responses in the liver [42].

The kidney is also the main organ attacked by AFB_1_. The renal toxicity of dietary AFB_1_ in broilers presented increased glomerular basement membrane thickening and stromal cells, glomerular enlargement, tubular epithelial cell cytoplasmic vacoulation, renal glomerulus collapse, and structural damage [43,44]. This study was not completely consistent with the aforementioned reports, possibly due to the different AFB_1_ concentrations in the diet.

Generally, the expression of Caspase-3 can promote apoptosis by inhibiting DNA repair and initiating DNA degradation, which can be activated by Caspase-8, Caspase-9, or mitochondrial cytochrome C. A similar study showed that the expressions of Capase-3 and Bax were increased in the thymus and bursa of Fabricius in broilers fed with AFB_1_, and the expression levels of Capase-3 and Bax were increased gradually with the increase of AFB_1_ concentration [45]. Based on the results of previous studies and this experiment, it can be inferred that the intestinal, liver, and kidney cells of broilers regulate inflammatory factors through an NOD-independent manner, which mainly activates the NF-κB inflammation signaling pathway by TNF-α and TLR factors. As an early inflammatory response, the expression of pattern receptors may be directly related to the concentration and action time of mycotoxin.

A previous study showed that *Lactobacillus* supplementa in broiler diets could suppress the LPS-induced expressions of pro-inflammatory genes (TNF-α, IL-1β, IL-6, IL-17, and IL-8) and improve the expressions of anti-inflammatory genes (IL-10 and TGF-β) in jejunum [13]. In the present study, the decreased inflammation of the liver and kidney after the addition of CMD confirmed the immune regulation function of probiotic metabolites. Several studies have confirmed that *Lactobacillus* can regulate the NF-κB signaling pathway and reduce the expression of pro-inflammatory factors by reducing the expressions of TLR2 and TLR4 [46,47]. It was found that the addition of yeast in piglet diets during *E. coli* challenges can significantly down-regulate the expressions of TLR2, TLR4, and pro-inflammatory factors in blood [48]. The extracellular polysaccharide produced by *Lactobacillus plantarum* up-regulated the expression of iNOS in macrophages by activating the NF-κB signaling pathway and induced the production of cytokines [49]. It was inferred that CMD alleviated AFB_1_-induced inflammatory response through NF-κB and TN-α signaling pathways via TLR pattern receptors in the intestinal, liver, and kidney tissues.

In the control group, *Lactobacillus* had the most abundance, followed by *Corynebacterium*, *Burkholderia-Caballeronia-Paraburkholderia*, and *Romboutsia*, which were not completely consistent with the previous study [50]. It may be related to some reasons such as sampling site, feed formula, and animal age. The previous study showed that AFB_1_ significantly reduced *Lactobacillus* abundance in the jejunum of broilers, which was returned to the normal level after the addition of compound probiotics including *Bacillus subtilis*, *Lactobacillus casein*, and *Candida utili* [16], in agreement with this study. In this study, AFB_1_ significantly increased the abundances of Staphylococcus and Escherichia-Shigella in addition to reducing the abundance of *Lactobacillus*.

Some studies have shown that probiotic supplementation in broiler diets can directly affect intestinal microbiota. The addition of *Enterococcus faecium* to broiler diets can reduce the fecal abundance of Clostridium perfringens and increase the level of potentially beneficial *Bifidobacteria* and *lactobacilli* [51]. *Lactobacillus casein* has been reported to improve the balance of intestinal microflora and digestive enzyme activities, as well as promote the growth of intestinal villi for broilers [25]. *Candida albicans* has the ability to adsorb AFB_1_ [52,53]. *Bacillus subtilis* BY2 can improve the innate immunity, disease resistance, and production performance of broilers [54]. A recent study found that the addition of *Bacillus subtilis* to broiler diets significantly increased the relative abundances of *Pseudomonas*, *Burkholderia*, and *Prevotella* [55], which was contrary to the results of this study. This may be due to the different probiotic composition and visible counts of microbes in the feed additives.

The microbe–host interactions have been a research hotspot in recent years, and several studies reported that the microbe–host interactions could be mediated through active mediators secreted by microbiota such as histamine, indole, and short-chain fatty acids [42,56]. PICRUSt (Phylogenetic Investigation of Communities by Reconstruction of Unobserved States) has been used to predict microbial community functions based on high-throughput sequencing results of prokaryotic 16S rRNA [57]. In this experiment, three kinds of kinases based on PICRUSt function prediction were significantly changed. IRAk-1 (Interleukin-1 receptor-associated kinase) is a key kinase in the TLR (Toll-like receptor) signaling pathway. After activation of TLRs, both IRAK-1 and IRAK-4 could be recruited by MyD88 to the subunits of the TLR receptor complex to cause IRAK-1 phosphorylation. The phosphorylated IRAK1 is released from the receptor complex, and the remaining receptor complex binds to TAK1 (TGF-β-activating kinase1) to form a new complex, which activates the inflammation-related signaling pathways through a complex series of phosphorylation reactions [58]. In this experiment, addition of AFB_1_ in the basal diet significantly improved the enrichment of TRAK-1, while CMD addition reduced the enrichment of TRAK-1. This result provides a support for the relationship between intestinal microbes and their regulating inflammatory responses.

JAK (Janus tyrosine kinase) belongs to a family of non-receptor protein tyrosine kinases, comprising of JAK1, JAK2, JAK3, and TYK2 (non-receptor protein tyrosine kinase-2). The JNK signaling pathway mediated caspase-dependent programmed cell death machinery through IRE1α (Inositol-requiring enzyme-1α) and ERK (extracellular regulated protein kinases) signaling molecules [10]. It was reported that AFB_1_ exposure could substantially increase endogenous ROS, which could activate the IRE1α/ERK1/2 signaling pathway and induce cell apoptosis. In this study, the enrichment of JAK was significantly reduced by AFB_1_, which may be due to the dose-dependency of the effects of ROS on activation of the JNK pathway. MEKK is a key kinase in the TNF-α mediated NF-κB signaling pathway [59]. It was reported that AFB_1_ induced oxidative stress and immunotoxicity via the phosphorylation of the ERK1/2 MAPK signal pathway in porcine alveolar macrophages [60]. In this study, the AFB_1_ significantly increased MEKK kinase enrichment, which suggests that AFB_1_ aggravates the inflammatory response through MAPK and NF-κB inflammatory pathways through TLR pattern receptors, while CMD significantly alleviated the inflammatory response, which was consistent with the qRT-PCR and protein expression results.

## 4. Conclusions

The mechanism of CMD alleviating AFB_1_-induced inflammatory responses in the intestine, liver and kidney tissues of broilers can be summarized as follows: (1) The degradation of AFB_1_ by CMD reduced the absorption and residue in broilers to decrease inflammation and tissue damage; (2) This study demonstrated that AFB_1_ up-regulates the expression of inflammatory cytokines through the TLR/NF-κB pathway, and CMD alleviates the inflammatory response by reducing the expression of inflammatory cytokines; (3) The study on the relationship between intestinal microbiota and inflammation based on the PICRUSt function prediction showed that CMD addition could keep gut microbiota stable, alter the enrichment of kinases related to the inflammatory pathways, and reduce AFB_1_ toxicity for broilers.

## 5. Materials and Methods

### 5.1. Compound Mycotoxin Detoxifier (CMD) Preparation

*Aspergillus oryzae*, *Lactobacillus casein*, *Bacillus subtilis*, *Candida utilis*, and *Enterococcus faecalis* were purchased from China General Microbiological Culture Collection Center, Beijing, China (CGMCC). *Bacillus subtilis* was inoculated in LB medium (g/L): peptone 10 g, yeast extract 5 g, NaCl 10 g, pH 7.0, and cultured in a rotary shaker with 180 rounds per min (rpm) at 37 °C for 24 h. *Lactobacillus casein* and *Enterococcus faecalis* were inoculated in MRS medium (g/L): peptone 10 g, yeast extract 10 g, glucose 20 g, Tween 80 1 mL, K_2_HPO_4_ 2 g, sodium acetate 5 g, sodium citrate 2 g, MgSO_4_ 0.2 g, MnSO_4_ 0.05 g, pH 6.20–6.60, cultured statically at 37 °C for 24 h. *Candida utilis* was inoculated in YPD medium (g/L): yeast extract 10 g, peptone 20 g, glucose 20 g, cultured at 30 °C for 24 h in 180 rpm shaker. After incubation, four species of microbes were placed statically for 2 h, and then the supernatant was removed. Skimmed milk powder, trehalose dihydrate, sodium glutamate, and silica were added and mixed for freeze-drying. The microbial counts were expressed as colony forming units per gram (CFU/g).

AFB_1_-degrading enzyme (ADE) was prepared from *Aspergillus oryzae*. *A. oryzae* incubation was prepared as follows: *A. oryzae* spores were scraped off from the incubating plate with sterilized normal saline and its concentration was adjusted to 1 × 10^8^ spores/mL. The solid-state medium formula was as follows(*w*/*w*): the ratio of wheat bran, corn meal, and soybean meal were 7:1:2, 15 g sample was taken, mixed with 9 mL distilled water, put in a 250 mL triangle bottle, autoclaved at 121 °C for 30 min, and then cooled to room temperature. The medium was inoculated with 2 mL of the aforementioned spore fluid, incubated at 30 °C for 5 days, and then dried. The activity of the AFB_1_-degrading enzyme was 1467 U/g. Enzyme activity was defined as the following: the amount of enzyme that could degrade 1 ng AFB_1_ per min at pH 8.0 and 37 °C was defined as one unit. One kilogram of CMD consisted of 667 g aflatoxin B_1_-degrading enzyme (ADE), 200 g montmorillonite, and 134 g compound probiotics (CP) in which the visible counts of *Bacillus subtilis*, *Lactobacillus casein*, *Enterococcus faecalis*, and *Candida utilis* were 1.0 × 10^8^, 1.0 × 10^8^, 1.0 × 10^10^, and 1.0 × 10^8^ CFU/g, respectively [22]. Montmorillonite was provided by Henan Delin Biological Product Co., Ltd. Xinxiang, Henan province, China.

### 5.2. Moldy Corn Collection and AFB_1_ Determination

The moldy corn was purchased from the market in Henan Province, China. AFB_1_ concentration in moldy corn was 70 μg/kg. According to the calculation of dietary formulation for broilers, all normal corn meals in the basal diet were replaced by the moldy corn meals to make the diets have high-level AFB_1_, in which the dietary AFB_1_ concentration was determined as 40 μg/kg. AFB_1_ concentration was measured by enzyme-linked immunosorbent assay with ELISA-RIDASCREEN AFB1 30/15 test kit (R-Biopharm, Darmstadt, Germany) according to the manufacturer’s standard instructions.

### 5.3. Animals and Managements 

The feeding experiment was divided into 2 stages (1–21 day and 22–42 day). In the early stage, a total of 200 one-day-old Ross broilers were randomly divided into 4 groups, 5 replications for each group, 10 broilers (half male and half female) in each replication. In the later feeding stage, 160 broilers at the age of 22 days were assigned to 4 groups with 8 replications in each group, and 5 broilers for each replication. The basal diet (group A) was prepared according to the recommended standard of nutrient requirement for broilers (NRC, 1994). The diets in Groups B and D were prepared by replacing the normal corn meals in the basal diet with moldy corn meals. All animals used in this experiment were managed according to the guidelines of Animal Care and Use Ethics Committee in Henan Agricultural University, Zhengzhou, Henan province, China (SKLAB-B-2010-003-01). The broilers were fed in multi-layer cages with 24 h light and natural ventilation. Feed and water were given to the birds *ad libitum*. The feeding experiment was designed as follows:Group A: Basal diet (4.31 μg/kg AFB_1_)Group B: Basal diet with moldy corn meal (40 μg/kg AFB_1_)Group C: Group A plus 1.5 g/kg CMDGroup D: Group B plus 1.5 g/kg CMD

### 5.4. Tissue Collection

After the 42-day feeding experiment, 4 broilers in each group were selected for slaughter sampling to collect jejunum, liver, and kidney tissues. The same parts of tissues were cut at 2 cm^3^, rinsed with normal saline, water-absorbed with filter paper, and then placed in 10% formalin fixative for further immunohistochemistry analysis. In addition, jejunum, liver, and kidney tissues were respectively placed in 2 mL sterilized cryotubes, quickly transferred to liquid nitrogen, and then transferred to a –80 °C refrigerator for quantitative gene and protein expression analyses of inflammation-related factors.

### 5.5. qRT-PCR Analysis

The total RNA was extracted from the intestine, liver, and kidney tissues of broilers using Trizol (Invitrogen, Carlsbad, CA, USA) according to the standard manufacturer’s instructions, and then dissolved in 50 μL RNase-free water and stored at −80 °C. The quality and concentration of RNA samples were measured by NanoDrop ND-1000 Spectrophotometer (Nano-Drop Technologies, Wilmington, DE, USA). Approximately 1 μg total RNA from each sample was reversely transcribed into cDNA by TB GREEN kit (TaKaRa, Dalian, China). Quantitative RT-PCR (qRT-PCR) was performed by the CFX Connect PCR Detection System (Bio-Rad, Hercules, CA, USA). All the primers used in this study are listed in Table 5. The β-actin was used as a house-keeping gene, and the relative mRNA abundances were analyzed using the 2^−ΔΔCT^ method [61].

### 5.6. Immunohistochemical (IHC) Staining

The formalin-fixed and paraffin-embedded tissues were subjected to IHC staining. Before staining, all specimens were incubated at 60 °C for 2 h. Then, the samples were deparaffinized in xylene, rehydrated in graded alcohol baths, and incubated in 3% H_2_O_2_ for 20 min. Heat-induced antigen retrieval was performed in 0.01 M sodium citrate buffer at pH 6.0. The primary antibody was reconstituted in 200 μL sterile PBS (0.5 mg/mL). Before the antibody incubation, each slide was incubated in 5% bull serum albumin for 15 min with the primary antibody (1:100, 50 μL/slide) for the experimental group and PBS for the control group overnight at 4 °C. Then, the slides were incubated with the secondary antibody for 30 min at 37 °C, stained with DAB (3,3′-diaminobenzidine) according to the protocol of the primary antibody, and counterstained with haematoxylin.

The result of cumulative optical density (COD) was quantified by Image J software (https://imagej.nih.gov/ij/) (accessed on 1 November 2020). Five fields from each slide were selected for the positive ratio of cell (PRC) calculation. Normal cells were stained in blue, and positive cells were stained on a spectrum from yellow to black. The size of the visual field was fixed at 50 × 50 μm (360 × 360 pixel), PRC = the total number of positive cells in 5 visual fields/the total number of cells in 5 visual fields. The results of PRC were divided into 4 grades: grade 1, negative (PRC ≤ 5); grade 2, weakly positive (5 < PRC ≤ 20); grade 3, positive (20 < PRC ≤ 50); grade 4, strong positive (50 < PRC ≤ 100).

### 5.7. Western Blotting (WB) Analysis

The total protein was extracted from each liver sample with RIPA lysis buffer. Briefly, protein samples were separated by 12% SDS-polyacrylamide gels and transferred to the PVDF membranes, blocked in 5% skimmed milk for 2 h, and subsequently incubated with primary antibody overnight at 4 °C. Rabbit polyclonal antibodies such as NF-κB (abs152602), NOD1 (abs135898), and TLR2 (abs136522) were purchased from Absin Bioscience Inc. (Shanghai, China). The β-actin was purchased from Bioworld Technology Inc. (Nanjing, China). Antibodies against NF-κB, TLR2, NOD1, and β-actin were diluted with 1:1000. After the membranes were washed 3 times for 10 min with tris buffered saline tween, they were incubated with secondary antibody for 2 h at room temperature. Finally, results were visualized by the chemiluminescence method and quantified by Image J software (V1.8.0.112).

### 5.8. Gut microbial Community Influenced by CMD and AFB_1_

The jejunum contents of broilers were used for 16S rRNA sequencing. The related sequencing was conducted by Shanghai Meji Biological Engineering (Shanghai, China). In details, DNA was extracted by the E.Z.N.A.^®^ soil DNA kit (Omega Bio-tek, Norcross, GA, USA). DNA concentration was quantified with a NanoDrop ND-2000 spectrophotometer (Thermo Fisher Scientific, Shanghai, China), and the quality was assessed with agarose gel electrophoresis.

The V3–V4 region of the 16S rRNA gene was amplified by polymerase chain reaction (PCR) with universal primers 338F (5′-ACTCCTACGGGAGGCAGCAG-3′) and 806R (5′-GGACTACHVGGGTWTCTAAT-3′). Specifically, sequencing adapters and barcodes were added to the 5′ end of universal primers for PCR. The purified amplifications were pooled in equimolar and paired-end sequenced on an Illumina platform (Biomarker Technology Co., Ltd., Beijing, China).

Raw fastq reads were demultiplexed into Trimmomatic [62] and merged by Pandaseq [63]. The merged reads were filtered according to the following criteria: (i) low-quality reads were removed when its average quality score was <30, and if ambiguous N bases were present, and the length of reads not between 220 bp and 500 bp; (ii) clean reads were ordered according to their abundance, and the singletons potentially generated by sequencing errors were also discarded. Operational taxonomic units (OTUs) were clustered at a cut-off of 97% sequence similarity with USEARCH (v7.1). The clean reads were aligned to the OTUs to obtain mapped reads for following analyses. For each representative sequence, the highest abundance sequence in each OTU was assigned taxonomies by RDP Classifier (http://rdp.cme.msu.edu/, accessed on 2 November 2020), against the silva database at an 70% confidence level. Subsequent bioinformatic analyses were conducted with the QIIME software package [64].

Based on 16S rRNA data, the Phylogenetic Investigation of Communities by Reconstruction of Unobserved States (PICRUSt 1.1.0) was used to generate a functional profile. 16S rRNA functional prediction was a method that uses PICRUSt to predict the functional composition of metagenome by marker gene data and a database of reference genomes such as the Cluster of Orthologous Groups (COG) and Kyoto Encyclopedia of Genes and Genomes (KEGG) databases.

### 5.9. Statistical Analysis

Data were presented as means ± standard deviations (SD) and were analyzed using one-way analysis of variance (ANOVA) by the Duncan method with SPSS 20.0 software (Sishu Software, Shanghai Co., Ltd., Shanghai, China). All graphs were generated using GraphPad Prism 8. Statistical significance was considered as *p*-value < 0.05.

## Figures and Tables

**Figure 1 toxins-14-00665-f001:**
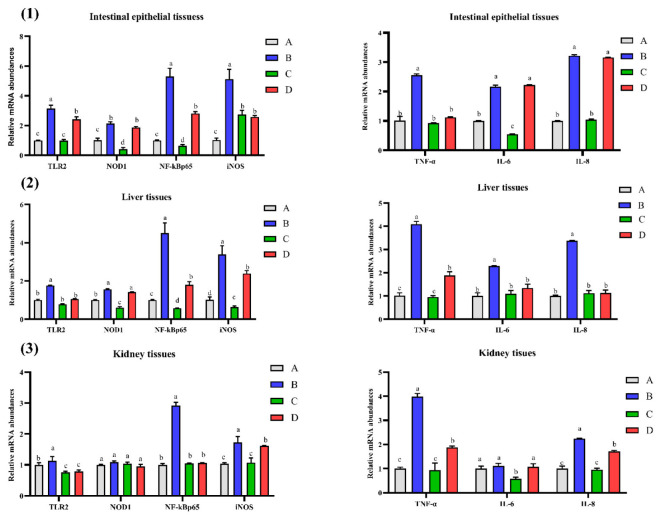
Effect of CMD on mRNA abundances of some genes in intestinal (**1**), liver (**2**), and kidney tissues (**3**) of broilers. Note: Group A: Basal diet (4.31 μg/kg AFB_1_); Group B: Basal diet with moldy corn meal (40 μg/kg AFB_1_); Group C: Group A plus 1.5 g/kg CMD; Group D: Group B plus 1.5 g/kg CMD. On each bar, significant differences at *p* < 0.05 levels are indicated by the different lowercase letters (a, b, c and d), while insignificant differences at *p* > 0.05 levels are indicated by the same lowercase letters.

**Figure 2 toxins-14-00665-f002:**
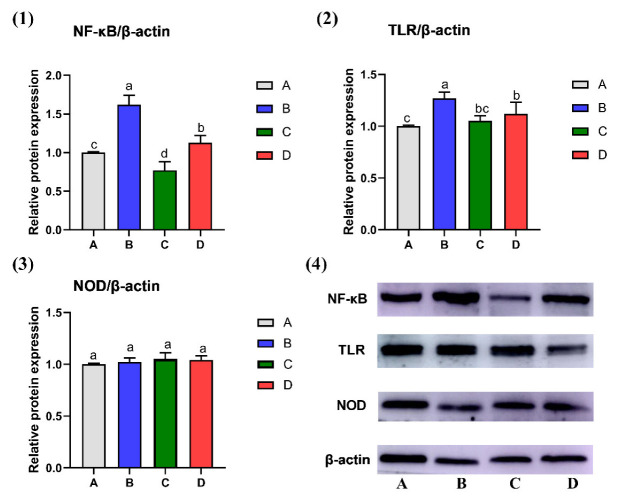
Effect of CMD on expression levels of NF-κB (**1**), TLR (**2**) and NOD (**3**) in liver tissue by WB analysis. Note: (**4**): electrophoretic band; Group A: Basal diet (4.31 μg/kg AFB_1_); Group B: Basal diet with moldy corn meal (40 μg/kg AFB_1_); Group C: Group A plus 1.5 g/kg CMD; Group D: Group B plus 1.5 g/kg CMD. On each bar, significant differences at *p* < 0.05 levels are indicated by the different lowercase letters (a, b, c, and d), while insignificant differences at *p* > 0.05 levels are indicated by the same lowercase letters.

**Figure 3 toxins-14-00665-f003:**
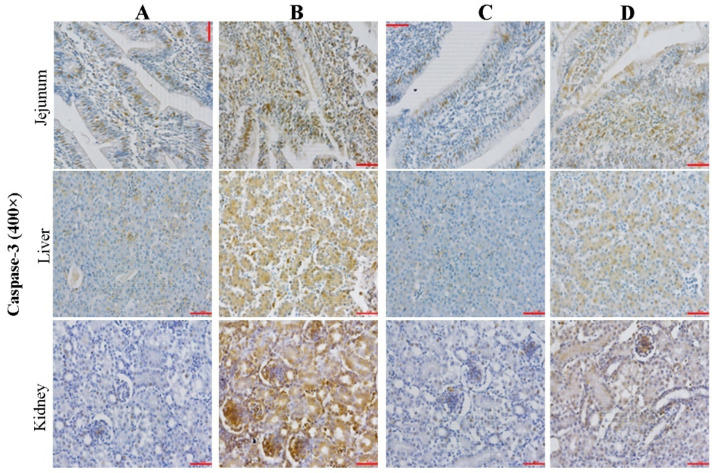
Effect of CMD on expression levels of Caspase-3 in intestinal, liver and kidney tissues by IHC analysis. Note: Group A: Basal diet (4.31 μg/kg AFB_1_); Group B: Basal diet with moldy corn meal (40 μg/kg AFB_1_); Group C: Group A plus 1.5 g/kg CMD; Group D: Group B plus 1.5 g/kg CMD. Bar = 50 μm.

**Figure 4 toxins-14-00665-f004:**
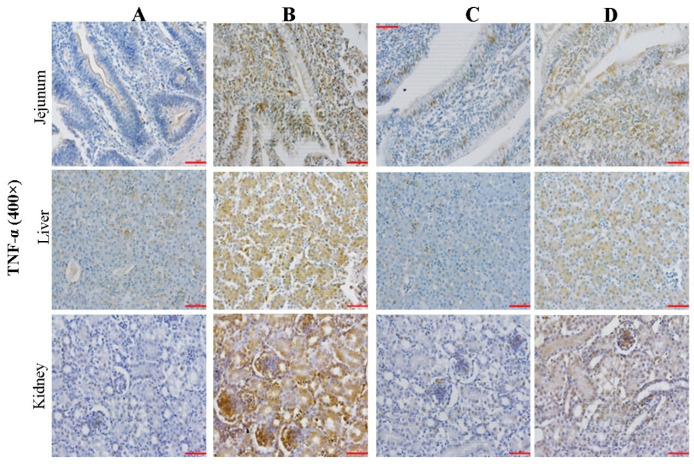
Effect of CMD on expression levels of TNF-α in intestinal, liver and kidney tissues. Note: Group A: Basal diet (4.31 μg/kg AFB_1_); Group B: Basal diet with moldy corn meal (40 μg/kg AFB_1_); Group C: Group A plus 1.5 g/kg CMD; Group D: Group B plus 1.5 g/kg CMD. Bar = 50 μm.

**Figure 5 toxins-14-00665-f005:**
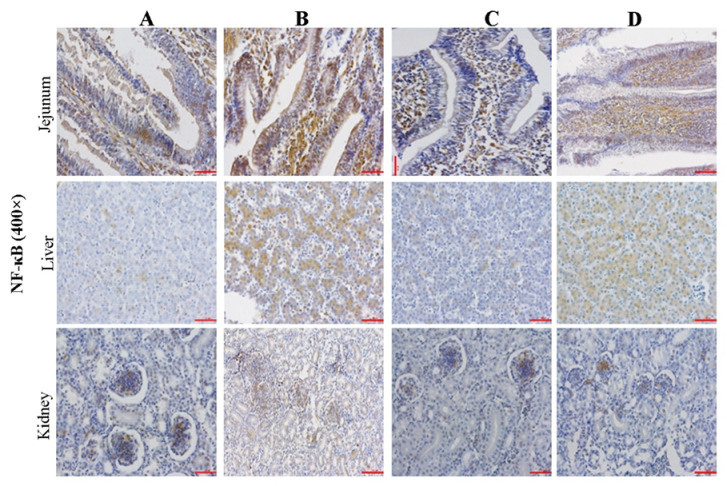
Effect of CMD on expression levels of TNF-α in intestinal, liver and kidney tissues. Note: Group A: Basal diet (4.31 μg/kg AFB_1_); Group B: Basal diet with moldy corn meal (40 μg/kg AFB_1_); Group C: Group A plus 1.5 g/kg CMD; Group D: Group B plus 1.5 g/kg CMD. Bar = 50 μm.

**Figure 6 toxins-14-00665-f006:**
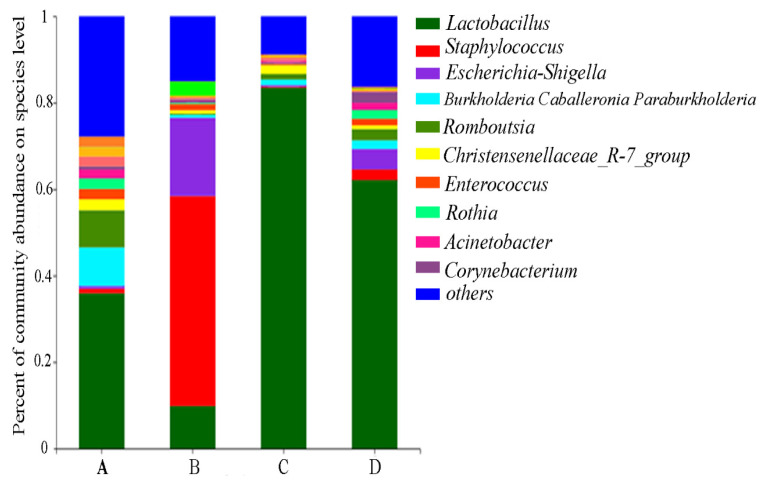
Relative abundances of microbiota in broiler jejunum at genus level. Note: Group A: Basal diet (4.31 μg/kg AFB_1_); Group B: Basal diet with moldy corn meal (40 μg/kg AFB_1_); Group C: Group A plus 1.5 g/kg CMD; Group D: Group B plus 1.5 g/kg CMD.

**Table 1 toxins-14-00665-t001:** The PRC and COD of Caspase-3 protein expressions in the intestine, liver, and kidney.

Groups	A	B	C	D
**PRC (%)**				
Intestine	27.73 ± 1.01 ^A,b^	50.57 ± 2.06 ^B,a^	19.35 ± 2.65 ^A,c^	22.14 ± 2.1 ^A,bc^
Liver	11.71 ± 1.52 ^B,b^	49.25 ± 2.4 ^B,a^	10.44 ± 1.95 ^B,b^	13.72 ± 1.24 ^B,b^
Kidney	3.51 ± 2.48 ^C,c^	69.87 ± 1.33 ^A,a^	25.24 ± 3.92 ^A,b^	26.12 ± 4.05 ^A,b^
**COD**				
Intestine	17.71 ± 1.4 ^A,b^	20.37 ± 1.04 ^C,a^	11.87 ± 2.27 ^A,c^	7.14 ± 0.62 ^C,d^
Liver	10.68 ± 0.37 ^B,c^	143.2 ± 14.18 ^A,a^	9.94 ± 0.7 ^A,c^	43.86 ± 3.92 ^A,b^
Kidney	7.60 ± 0.66 ^C,c^	85.04 ± 5.67 ^B,a^	5.27 ± 0.34 ^B,c^	28.1 ± 4.08 ^B,b^

Note: In the same column, significant differences at *p* < 0.05 levels are indicated by different capital letters (A, B, and C), while insignificant differences at *p* > 0.05 levels are indicated by the same capital letters. In the same row, significant differences at *p* < 0.05 levels are indicated by different lowercase letters (a, b, c, and d), while insignificant differences at *p* > 0.05 levels are indicated by the same lowercase letters.

**Table 2 toxins-14-00665-t002:** The PRC and COD of TNF-α expressions in jejunum, liver, and kidney of broilers.

Groups	A	B	C	D
**PRC (%)**				
Intestine	0.02 ± 0.01 ^A,c^	26.19 ± 2.3 ^B,a^	8.86 ± 3.07 ^A,b^	7.15 ± 1.71 ^C,b^
Liver	1.36 ± 1.39 ^A,c^	32.1 ± 3.37 ^A,a^	0.00 ± 0.00 ^B,c^	11.53 ± 2.8 ^B,b^
Kidney	0.00 ± 0.00 ^A,c^	23.73 ± 2.15 ^B,a^	3.27 ± 2.33 ^A,b^	19.2 ± 2.93 ^A,a^
**COD**				
Intestine	7.60 ± 0.71 ^C,c^	79.12 ± 4.18 ^B,a^	7.18 ± 0.21 ^A,c^	26.48 ± 3.83 ^B,b^
Liver	18.06 ± 1.07 ^A,d^	124.67 ± 9.33 ^A,a^	3.07 ± 0.73 ^B,c^	60.99 ± 4.55 ^A,b^
Kidney	9.45 ± 0.93 ^B,c^	34.24 ± 2.62 ^C,a^	4.4 ± 0.28 ^B,d^	12.01 ± 1.51 ^C,b^

Note: In the same column, significant differences at *p* < 0.05 levels are indicated by different capital letters (A, B, and C), while insignificant differences at *p* > 0.05 levels are indicated by the same capital letters. In the same row, significant differences at *p* < 0.05 levels are indicated by different lowercase letters (a, b, c and d), while insignificant differences at *p* > 0.05 levels are indicated by the same lowercase letters.

**Table 3 toxins-14-00665-t003:** The PRC and COD of NF-κB expressions in jejunum, liver, and kidney of broilers.

Groups	A	B	C	D
**PRC (%)**				
Intestine	20.27 ± 3.06 ^A,b^	83.63 ± 2.84 ^A,a^	6.47 ± 1.76 ^A,c^	25.04 ± 2.55 ^A,b^
Liver	1.08 ± 1.52 ^B,b^	13.52 ± 2.96 ^B,a^	0.00 ± 0.00 ^B,b^	7.8 ± 1.82 ^B,a^
Kidney	1.23 ± 1.75 ^B,b^	13.51 ± 1.85 ^B,a^	1.33 ± 1.89 ^B,b^	2.62 ± 1.86 ^C,b^
**COD**				
Intestine	15.8 ± 1.21 ^A,b^	27.12 ± 0.65 ^B,a^	15.57 ± 0.57 ^A,b^	15.67 ± 0.33 ^B,b^
Liver	11.7 ± 2.26 ^B,c^	39.39 ± 5.64 ^A,a^	13.2 ± 0.81 ^B,c^	23.25 ± 2.51 ^A,b^
Kidney	11.68 ± 1.84 ^B,b^	35.55 ± 4.05 ^A,a^	13.18 ± 1.78 ^B,b^	7.56 ± 1.24 ^C,c^

Note: In the same column, significant differences at *p* < 0.05 levels are indicated by different capital letters (A, B, and C), while insignificant differences at *p* > 0.05 levels are indicated by the same capital letters. In the same row, significant differences at *p* < 0.05 levels are indicated by different lowercase letters (a, b, and c), while insignificant differences at *p* > 0.05 levels are indicated by the same lowercase letters.

**Table 4 toxins-14-00665-t004:** The kinase related to the inflammatory pathway based on PICRUSt function prediction.

Groups		A	B	C	D
EC 2.7.11.1	Interleukin-1 receptor-associated kinase (IRAK-1)	42,321 ± 385 ^b^	686,817 ± 416 ^a^	27,456 ± 321 ^c^	47,551 ± 316 ^b^
EC 2.7.10.2	Janus tyrosine Kinase (JAK)	212 ± 1.78 ^b^	57.21 ± 1.32 ^c^	223.38 ± 12.36 ^a^	211.87 ± 10.1 ^a^
EC2.7.11.25	Mitogen-activated protein kinase (MAPK)	25,357 ± 417 ^a^	30,569 ± 396 ^a^	14,532 ± 677 ^b^	27,651 ± 386 ^a^

Note: In the same row, significant differences at *p* < 0.05 levels are indicated by different lowercase letters (a, b, and c), while insignificant differences at *p* > 0.05 levels are indicated by the same lowercase letters.

**Table 5 toxins-14-00665-t005:** Primer sequences of some genes for quantitative RT-PCR.

Gene	Accession Number	Primer Sequence (5′–3′)
β-actin	LO8165	F: GAGAAATTGTGCGTGACATCA
		R: CCTGAACCTCTCATTGCCA
IL-6	AJ309540	F: CAAGGTGACGGAGGAGGAC
		R: TGGCGAGGAGGGATTTCT
IL-8	AJ009800	F: ATGAACGGCAAGCTTGGAGCTG
		R: TCCAAGCACACCTCTCTTCCATCC
iNOS	U46504	F: CAGCTGATTGGGTGTGGAT
		R: TTTCTTTGGCCTACGGGTC
NF-κBp65	NM_205129	F: GTGTGAAGAAACGGGAACTG
		R: GGCACGGTTGTCATAGATGG
TNF-α	NM_204267	F: GAGCGTTGACTTGGCTGTC
		R: AAGCAACAACCAGCTATGCAC
NOD1	JX465487	F: AGCACTGTCCATCCTCTGTCC
		R: TGAGGGTTGGTAAAGGTCTGCT
TLR2	NM_001161650	F: GGGGCTCAGGCAAAATC
		R: AGCAGGGTTCTCAGGTTCACA

## Data Availability

Not applicable.

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
