# Peer review of "Effects of Compound Mycotoxin Detoxifier on Alleviating Aflatoxin B1-Induced Inflammatory Responses in Intestine, Liver and Kidney of Broilers"

_toxins, 2022, doi:10.3390/toxins14100665_

Round 1
Reviewer 1 Report
The manuscript is well written, and I found it very interesting. However, I have few questions, which I want authors to answer before publication.
1. How CMD was administrated?
2. Does authors have any particular reason why they evaluated the protein expression of molecular markers only in liver not in other tissues?
3. Figure legends need more clarity in terms of significance indication. I couldn’t understand what group A was compared to?
4. In figure 2(2) what group c was compared with? Why it has tow different significance in it?
5. Similarly, in figure 2(3), are they even significantly different from each other?
6. How CMD helps to inhibits the toxic effects of AFB1, what it’s mechanism of action?
7. What was the length and weight of the organ to body weight ratio? It is very important to analyze as it will help to conclude the gross toxicity in the body with and without treatment.
8. Based of what evidence authors have decided to select the dosage of AFB1?
9. There are several typo errors in the whole manuscript and also in the abstract. Check it carefully before final submission.
10. Below mentioned articles are suitable for citation:
Dey DK, et al., Environ Pollut. 2021. PMID: 33038573.
Dey DK, et al., Crit Rev Food Sci Nutr. 2022. PMID: 35445609.
Zolfaghari H, et al., Adv Pharm Bull. 2020. PMID: 32665910.
Author Response
Response to the comments of reviewer 1
(All the revised parts have been marked with red color in the revised manuscript)
Comments 1: How CMD was administrated?
Response 1:
The compound mycotoxin detoxifier (CMD) preparation was as follows:
Aspergillus oryzae, Lactobacillus casein, Bacillus subtilis, Candida utilis and Enterococcus faecalis were purchased from China General Microbiological Culture Collection Center (CGMCC). Bacillus subtilis was inoculated in LB medium (g/L): peptone 10 g, yeast extract 5 g, NaCl 10 g, pH 7.0, and cultured in a rotary shaker with 180 rounds per min (rpm) at 37°C for 24 h. Lactobacillus casein and Enterococcus faecalis were inoculated in MRS medium (g/L): peptone 10 g, yeast extract 10 g, glucose 20 g, Tween 80 1 mL, K2HPO4 2 g, sodium acetate 5 g, sodium citrate 2 g, MgSO4 0.2 g, MnSO4 0.05 g, pH 6.20-6.60, cultured statically at 37°C for 24 h. Candida utilis was inoculated in YPD medium (g/L): yeast extract 10 g, peptone 20 g, glucose 20 g, cultured at 30°C for 24 h in 180 rpm shaker. After incubation, four species of microbes were placed statically for 2 h, and then the supernatant was removed. Skimmed milk powder, trehalose dihydrate, sodium glutamate, and silica were added and mixed for freeze-drying. The microbial counts were expressed as colony forming units per gram (CFU/g).
AFB1-degrading enzyme (ADE) was prepared from Aspergillus oryzae. A. oryzae incubation was prepared as follows: A. oryzae spores were scraped off from the incubating plate with sterilized normal saline, and its concentration was adjusted to 1×108 spores/mL. The solid-state medium formula was as the following(w/w): the ratio of wheat bran, corn meal and soybean meal were 7:1:2, 15 g sample was taken, mixed with 9 mL distilled water, put in a 250 mL triangle bottle, autoclaved at 121°C for 30 min, and then cooled to room temperature. The medium was inoculated with 2 mL of the above spore fluid, incubated at 30°C for 5 d, and then dried. The activity of AFB1-degrading enzyme was 1467 U/g. Enzyme activity was defined as the following: the amount of enzyme that could degrade 1 ng AFB1 per min at pH 8.0 and 37℃ was defined as one unit. One kilogram of CMD consisted of 667 g aflatoxin B1-degrading enzyme (ADE), 200 g montmorillonite, and 134 g compound probiotics (CP) in which the visible counts of Bacillus subtilis, Lactobacillus casein, Enterococcus faecalis and Candida utilis were 1.0×108, 1.0×108, 1.0×1010, and 1.0×108 CFU/g, respectively [22]. Montmorillonite was provided by Henan Delin Biological Product Co., Ltd. Xinxiang, China.
The above information has been added in “Materials and methods” section in the revised manuscript.
Comments 2: Does authors have any particular reason why they evaluated the protein expression of molecular markers only in liver not in other tissues?
Response 2:
The liver was chosen for two main reasons:
- Liver is the main organ attacked by AFB1. Compare with other cells, liver cells are the most sensitive to AFB1.
- In our previous study (Guo, H. W., et al. Effects of compound probiotics and aflatoxin-degradation enzyme on alleviating aflatoxin-induced cytotoxicity in chicken embryo primary intestinal epithelium, liver and kidney cells. AMB Express. 2021, 11(1), 35. https://doi.org/10.1186/s13568-021-01196-7), AFB1 could induce inflammatory responses in chick embryo intestine, liver and kidney cells through suppressing the activations of NF-κB, iNOS, NOD1 and TLR2 pathways, and these indexes showed the same change trend among three kinds of cells.
Therefore, we evaluated the protein expression of molecular markers only in liver and not in the other tissues.
Comments 3: Figure legends need more clarity in terms of significance indication. I couldn’t understand what group A was compared to?
Response 3:
The comparisons were conducted among four groups. On each bar, significant differences at P<0.05 levels are indicated by the different lowercase letters (a, b, c and d), while insignificant differences at P>0.05 levels are indicated by the same lowercase letters.
Comments 4: In figure 2(2) what group c was compared with? Why it has two different significance in it?
Response 4:
In figure 2(2), it was insignificant different between group C and group D or between group C and group A, but it was significant different between group A and group D; therefore, it is correct. On each bar, significant differences at P<0.05 levels are indicated by the different lowercase letters (a, b, c, and d), while insignificant differences at P>0.05 levels are indicated by the same lowercase letters.
Comments 5: Similarly, in figure 2(3), are they even significantly different from each other?
Response 5:
After statistical analysis, there was no significant difference among four groups (P>0.05).
Comments 6: How CMD helps to inhibits the toxic effects of AFB1, what it’s mechanism of action?
Response 6:
The mechanism of CMD alleviating AFB1-induced inflammatory responses in the intestine, liver and kidney tissues of broilers can be summarized as follows:
1) The degradation of AFB1 by CMD reduced the absorption and residue in broilers
2) This study demonstrated that AFB1 up-regulates the expression of inflammatory cytokines through the TLR/NF-κB pathway, and CMD alleviates the inflammatory response by reducing the expression of inflammatory cytokines;
3) CMD could keep gut microbiota stable, alter the enrichment of kinases related with the inflammatory pathways, and reduce AFB1 toxicity for broilers.
Comments 7: What was the length and weight of the organ to body weight ratio? It is very important to analyze as it will help to conclude the gross toxicity in the body with and without treatment.
Response 7:
Organ index, blood biochemical index and tissue residue were all good indicators to evaluate the detoxification effect of CMD. But they were not reflected in this study due to too much content. Actually, there was no significant change in organ index of intestine, liver and kidney tissues, but significant change in organ index bursa of Fabricius and thymus. At present, we are further in-depth research on bursa of Fabricius and thymustissues tissues, these interesting results are prepared for another manuscript.
Comments 8: Based of what evidence authors have decided to select the dosage of AFB1?
Response 8:
In our previous study (Guo, H. W., et al. Detoxification of aflatoxin B1 in broiler chickens by a triple-action feed additive. Food Addit Contam A. 2021, 38(9), 1583-1593. https://doi.org/10.1080/19440049.2021.1957159), it has been proved that 40 μg/kg AFB1 could significantly inhibit the production of broilers. Therefore, 40 μg/kg AFB1 was selected as a negative
control.
Comments 9: There are several typo errors in the whole manuscript and also in the abstract. Check it carefully before final submission.
Response 9:
Thank you for your comments. It has been revised and marked with red color in the revised manuscript.
Comments 10: Below mentioned articles are suitable for citation:
Dey DK, et al., Environ Pollut. 2021. PMID: 33038573.
Dey DK, et al., Crit Rev Food Sci Nutr. 2022. PMID: 35445609.
Zolfaghari H, et al., Adv Pharm Bull. 2020. PMID: 32665910.
Response 10:
Thank you for your suggestion. These articles have been cited in the introduction and discussion sections in the revised manuscript.

Reviewer 2 Report
Figures 3-5 be improved for quality presentation. Suggested following references to digest in Introduction and/or Discussion:
Imran M, Cao S, Wan SF, Chen Z, Saleemi MK, Wang N, Naseem MN and Munawar J, 2020. Mycotoxins - a global one health concern: A review. Agrobiological Records 2: 1-16. https://doi.org/10.47278/journal.abr/2020.006
Abdel-Sattar WM, KM Sadek, AR Elbestawy, DM Mourad and HS El-Samahy, 2019. Immunological, histopathological and biochemical protective effect of date pits (Phoenix dacrylifera seeds) feed additive against aflatoxicated broiler chickens. International Journal of Veterinary Science 8: 198-205.
Ali, A.M.A., Fahmy, M.F., Metwally, M.M., (...), Azazy, H.A., Mowafy, R.E. 2021. Ameliorative effects of cholestyramine and oxihumate on aflatoxicosis in broiler chickens. Pakistan Veterinary Journal 41(1), pp. 51-56
Saleemi MK, MK Ashraf, ST Gul, MN Naseem, MS Sajid, M Mohsin, C He, M Zubair and A Khan, 2020. Toxicopathological effects of feeding aflatoxins B1 in broilers and its amelioration with indigenous mycotoxin binder. Ecotoxicology and Environmental Safety, 187: 109712. https://doi.org/10.1016/j.ecoenv.2019.109712
Ashraf A, Saleemi MK, Mohsin M, Gul ST, Zubair M, Muhammad F, Bhatti SA, Hameed MR, Imran M, Irshad H, Zaheer I, Ahmed I, Raza A, Qureshi AS and Khan A, 2022. Pathological effects of graded doses of aflatoxin B1 on the development of testes in juvenile white Leghorn males. Environmental Science and Pollution Research 29: 53158–53167. https://doi.org/10.1007/s11356-022-19324-6
Author Response
Response to the comments of reviewer 2
(All the revised parts have been marked with red color in the revised manuscript)
Comments 1:
Figures 3-5 be improved for quality presentation. Suggested following references to digest in Introduction and/or Discussion:
Imran M, Cao S, Wan SF, Chen Z, Saleemi MK, Wang N, Naseem MN and Munawar J, 2020. Mycotoxins - a global one health concern: A review. Agrobiological Records 2: 1-16.
Abdel-Sattar WM, KM Sadek, AR Elbestawy, DM Mourad and HS El-Samahy, 2019. Immunological, histopathological and biochemical protective effect of date pits (Phoenix dacrylifera seeds) feed additive against aflatoxicated broiler chickens. International Journal of Veterinary Science 8: 198-205.
Ali, A.M.A., Fahmy, M.F., Metwally, M.M., (...), Azazy, H.A., Mowafy, R.E. 2021. Ameliorative effects of cholestyramine and oxihumate on aflatoxicosis in broiler chickens. Pakistan Veterinary Journal 41(1), pp. 51-56
Saleemi MK, MK Ashraf, ST Gul, MN Naseem, MS Sajid, M Mohsin, C He, M Zubair and A Khan, 2020. Toxicopathological effects of feeding aflatoxins B1 in broilers and its amelioration with indigenous mycotoxin binder. Ecotoxicology and Environmental Safety, 187: 109712.
Ashraf A, Saleemi MK, Mohsin M, Gul ST, Zubair M, Muhammad F, Bhatti SA, Hameed MR, Imran M, Irshad H, Zaheer I, Ahmed I, Raza A, Qureshi AS and Khan A, 2022. Pathological effects of graded doses of aflatoxin B1 on the development of testes in juvenile white Leghorn males. Environmental Science and Pollution Research 29: 53158–53167.
Response 1:
Figures 3-5 quality can not be improved due to the technological problem. These articles have been cited in the introduction and discussion sections in the revised manuscript. See below for details:
According to the Figures 3-5, microscopic damages were clearly observed first. The damages of AFB1 to intestinal tract mainly present barrier function loss and inflammatory reaction, lymphocyte or monocyte infiltration as well as mucosal hyperplasia and vacuolar degeneration, and even decrease intestinal villus height and in-crease intestinal crypt depth [37,38]. In this study, disruption of the intestinal villi was clearly observed, while the disruption did not decrease with the addition of CMD. It suggesting that the AFB1-induced negative effect in villus was irreversible and CMD did not significantly reduce the direct intestinal damage caused by AFB1.
Liver is the primary organ attacked by AFB1 which can cause many microscopic damages including high-level eosinophile granulocyte and monocytes, lipid vacuoles [39], inflammatory cell proliferation and infiltration, the edema and hepatocytes de-generation [40], in agreement with this study. however, the CMD additions could alleviate mycotoxin negative effects on the Liver tissue damage. The results of this experiment show that the CMD could significantly alleviate the liver microcosmic damage caused by AFB1. It was reported that compound probiotics with aflatoxin B1-degrading enzyme could improve AFB1 metabolism, hepatic cell structure, antioxidant activity of broilers exposed to AFB1-contaminated diet [41]. Saeedi et al. demonstrate that the gut microbiome acts at a distance to activate host antioxidant responses in the liver [42].
Kidney is also the main organ attacked by AFB1. The renal toxicity of dietary AFB1 in broilers presented the increased glomerular basement membrane thickening and stromal cells, glomerular enlargement, tubular epithelial cell cytoplasmic vacoulation, renal glomerulus collapse, and structural damage [43,44]. Not completely consistent with previous studies, the glomerular morphology was basically complete with boundary, this may be due to the different AFB1 concentrations in the diet.

Reviewer 3 Report
the MS investigate in vitro the effects of CDM on mitigating AFB1 cytotoxicity in chicken embryo primary intestinal epithelium, liver and kidney cells. To further clarify the effectiveness of CMD in vivo, this study will explored the mechanism of this treatment in vivo. the study is well designed even if the composition of CDM is not report. authors state that it has been published but the Journal is not open access therefore they must to reoort what CDM is.
Author Response
Response to the comments of reviewer 3
(All the revised parts have been marked with red color in the revised manuscript)
Comments 1:
the MS investigate in vitro the effects of CDM on mitigating AFB1 cytotoxicity in chicken embryo primary intestinal epithelium, liver and kidney cells. To further clarify the effectiveness of CMD in vivo, this study will explore the mechanism of this treatment in vivo. the study is well designed even if the composition of CDM is not report. authors state that it has been published but the Journal is not open access therefore they must to report what CDM is.
Response 1:
Thanks for your comments. The compound mycotoxin detoxifier (CMD) preparation was as follows:
Aspergillus oryzae, Lactobacillus casein, Bacillus subtilis, Candida utilis and Enterococcus faecalis were purchased from China General Microbiological Culture Collection Center (CGMCC). Bacillus subtilis was inoculated in LB medium (g/L): peptone 10 g, yeast extract 5 g, NaCl 10 g, pH 7.0, and cultured in a rotary shaker with 180 rounds per min (rpm) at 37°C for 24 h. Lactobacillus casein and Enterococcus faecalis were inoculated in MRS medium (g/L): peptone 10 g, yeast extract 10 g, glucose 20 g, Tween 80 1 mL, K2HPO4 2 g, sodium acetate 5 g, sodium citrate 2 g, MgSO4 0.2 g, MnSO4 0.05 g, pH 6.20-6.60, cultured statically at 37°C for 24 h. Candida utilis was inoculated in YPD medium (g/L): yeast extract 10 g, peptone 20 g, glucose 20 g, cultured at 30°C for 24 h in 180 rpm shaker. After incubation, four species of microbes were placed statically for 2 h, and then the supernatant was removed. Skimmed milk powder, trehalose dihydrate, sodium glutamate, and silica were added and mixed for freeze-drying. The microbial counts were expressed as colony forming units per gram (CFU/g).
AFB1-degrading enzyme (ADE) was prepared from Aspergillus oryzae. A. oryzae incubation was prepared as follows: A. oryzae spores were scraped off from the incubating plate with sterilized normal saline, and its concentration was adjusted to 1×108 spores/mL. The solid-state medium formula was as the following(w/w): the ratio of wheat bran, corn meal and soybean meal were 7:1:2, 15 g sample was taken, mixed with 9 mL distilled water, put in a 250 mL triangle bottle, autoclaved at 121°C for 30 min, and then cooled to room temperature. The medium was inoculated with 2 mL of the above spore fluid, incubated at 30°C for 5 d, and then dried. The activity of AFB1-degrading enzyme was 1467 U/g. Enzyme activity was defined as the following: the amount of enzyme that could degrade 1 ng AFB1 per min at pH 8.0 and 37℃ was defined as one unit. One kilogram of CMD consisted of 667 g aflatoxin B1-degrading enzyme (ADE), 200 g montmorillonite, and 134 g compound probiotics (CP) in which the visible counts of Bacillus subtilis, Lactobacillus casein, Enterococcus faecalis and Candida utilis were 1.0×108, 1.0×108, 1.0×1010, and 1.0×108 CFU/g, respectively [22]. Montmorillonite was provided by Henan Delin Biological Product Co., Ltd. Xinxiang, China.
The above information has been added in “Materials and methods” section in the revised manuscript.

Reviewer 4 Report
Manuscript Number: toxins-1918952
Title: Effects of compound mycotoxin detoxifier on alleviating aflatoxin B1-induced inflammatory responses in intestine, liver and kidney of broilers
The submitted manuscript presents results of the study on the effectiveness of a triple-action compound mycotoxin detoxifier containing the aflatoxin B1-degrading enzyme, montmorillonite and compound probiotics on inflammatory response in intestine, liver and kidney of broilers in vivo. In addition, the authors try to explain the mechanism of the used detoxifier for alleviating inflammation responses in the intestine, liver and kidney of broilers caused by AFB1. The context of the work indicates that the presented considerations are a continuation of the authors' earlier research works, conducted in vitro and partly in vivo. In my opinion, the study is valuable extension of the knowledge related to the consequences of mycotoxin contamination and may be of interest to the readers of the journal. Additionally, the work has also an application nature, because mycotoxins in feed can easily penetrate into animal products and further into the human food chain. Thus, any method capable of preventing contamination of food products with metabolites of microscopic fungi is of great importance for food producers and for the safety and health of consumers. The authors correctly defined the research goal and supported its purposefulness with the analysis of the literature. The results are also clearly presented and discussed with previous research, but the experimental design and quality of the used feed material raises certain doubts.
The use of the diet containing naturally mycotoxin-contaminated ingredients (as opposed to an artificially fortified diet) is good idea, but requires thorough examination. The authors purchased moldy corn that could be contaminated with other fungal metabolites as well. Have the authors determined other mycotoxins in the maize? In the case of their presence, the observed changes in the tissues of the examined organs would be the result of the cumulative action of all these toxins. The alleviating effect of the applied detoxifier may also be different in the coexistence of several toxins than in the case of AFB1 alone.
The subsection 5.3. Animals and managements: It is advisable that the authors emphasize more clearly why the experiment was divided into two breeding periods. Moreover, in line 346 the authors wrote that: “During the experiment, body weight, feed intake and mortality were recorded, and average daily feed intake (ADFI) and average daily gain (ADG) were measured” but the Results and Discussion sections do not refer to it at all. Probably the experiment was broader and some of the results were described in other articles and this text is redundant. If so, it should be deleted, otherwise the relevant results should be completed in the manuscript.
Editorial mistakes:
Line 336: There is a capital letter in the middle of the sentence: “stage, A total”.
Line 376: The chemical notation of a compound (H2O2) requires the use of subscripts.
Line 384: There is redundant letter "T".
Author Response
Response to the comments of reviewer 4
(All the revised parts have been marked with red color in the revised manuscript)
Comments 1:
The use of the diet containing naturally mycotoxin-contaminated ingredients (as opposed to an artificially fortified diet) is good idea, but requires thorough examination. The authors purchased moldy corn that could be contaminated with other fungal metabolites as well. Have the authors determined other mycotoxins in the maize? In the case of their presence, the observed changes in the tissues of the examined organs would be the result of the cumulative action of all these toxins. The alleviating effect of the applied detoxifier may also be different in the coexistence of several toxins than in the case of AFB1 alone.
Response 1:
The AFB1, zearalenone (ZEA), and deoxynivalenol (DON) concentrations in moldy corn meal were 42.36, 102.28 and 138.06 μg/kg, respectively. Compared with 10, 1000 and 3000 μg/kg for AFB1, ZEA and DON maximum limit standard for broiler diet in China, only AFB1 content exceeds the limit. Therefore, this study focused on the toxicity of AFB1 to broilers.
Comments 2:
The subsection 5.3. Animals and managements: It is advisable that the authors emphasize more clearly why the experiment was divided into two breeding periods. Moreover, in line 346 the authors wrote that: “During the experiment, body weight, feed intake and mortality were recorded, and average daily feed intake (ADFI) and average daily gain (ADG) were measured” but the Results and Discussion sections do not refer to it at all. Probably the experiment was broader and some of the results were described in other articles and this text is redundant. If so, it should be deleted, otherwise the relevant results should be completed in the manuscript.
Response 2:
Thank you for your comments. The above information has been deleted because the related materials are prepared for another manuscript.
Comments 3:
Editorial mistakes:
Line 336: There is a capital letter in the middle of the sentence: “stage, A total”.
Line 376: The chemical notation of a compound (H2O2) requires the use of subscripts.
Line 384: There is redundant letter "T".
Response 3:
Thank you for your comments. They have been revised and marked with red color in the revised manuscript.
